# SAR2Earth: A SAR-to-EO Translation Dataset for Remote Sensing Applications

## Abstract

Electro-optical (EO) images are essential to a wide range of remote sensing applications. With the advent of data-driven models, the efficiency of EO image analysis has significantly improved, enabling faster and more effective outcomes in these applications. However, EO images have inherent limitations—they cannot penetrate cloud cover and are unable to capture imagery at night. To overcome these challenges, synthetic aperture radar (SAR) images are employed, as they can operate effectively regardless of weather conditions or time of day. Despite this advantage, SAR images come with their own difficulties: they are affected by speckle noise, complicating analysis, and existing algorithms developed for EO imagery are not directly transferable to SAR data. To address these issues, we introduce SAR2Earth, a benchmark dataset specifically designed for SAR-to-EO translation. By translating SAR images into EO-like representations, SAR2Earth allows the extensive range of algorithms developed for EO imagery to be applied effectively to SAR data. The dataset consists of 18 spatially aligned pairs of SAR and EO images, collected from 8 distinct regions encompassing both urban and rural. We provide comprehensive evaluations, detailed model analyses, and extensive experimental results. All codes and datasets will be made publicly available at `https://sar2earth.github.io`.

## 1 Introduction

Remote sensing images provide the capability to observe the Earth on a large scale, making them invaluable for analysis in various applications such as transportation (Ball et al., 2017), defense (Xu et al., 2024), natural resource management (Kumar et al., 2015), disaster response (AlAli & Alabady, 2022), and environmental monitoring. However, the vast amount of data generated poses significant challenges for manual analysis due to the time and expertise required. The advent of data-driven models (Wang et al., 2021; Oh et al., 2023; Kuckreja et al., 2024) has enabled more efficient and effective analysis of these images. Electro-optical (EO) imagery has been the primary modality for remote sensing applications due to its intuitive representation of the Earth. However, EO imagery has significant limitations: it cannot penetrate cloud cover and is unable to capture images at night, restricting its utility in many scenarios (Seo et al., 2023; Low et al., 2023b). For instance, during natural disasters like floods—which are often accompanied by heavy cloud cover—EO imagery becomes ineffective for timely disaster assessment and response. To overcome these limitations, synthetic aperture radar (SAR) imagery is employed. SAR sensors can operate independently of daylight and weather conditions, providing consistent imaging capabilities. However, SAR images suffer from speckle noise due to the coherent nature of radar signal processing, which introduces granular interference patterns. This speckle noise makes SAR images challenging to interpret (Spigai et al., 2011; Zhang et al., 2015), especially for non-experts, and complicates the application of algorithms developed for EO imagery. To bridge this gap, SAR-to-EO translation methods (Fuentes Reyes et al., 2019; Wang et al., 2022a; Yang et al., 2022; Lee et al., 2023) have been proposed, aiming to translate SAR images into EO-like images that are more accessible for analysis using existing EO-based algorithms.

Despite these efforts, there has been a lack of comprehensive analysis of these methods, and they often remain isolated applications without standardized benchmarks. Existing SAR and EO multimodal datasets (Schmitt et al., 2018; Wang & Zhu, 2018; Shermeyer et al., 2020; Low et al., 2023a; 2024) are limited in both quantity and diversity, often being captured within specific regions. This

Table 1: Comparison of SAR2Earth dataset with existing SAR and EO multi-modality datasets. P, MS, and PS represent Panchromatic (P), Multi-Spectral (MS), and Pan-Sharpened (PS) image types. Tasks (DF, BE, I2I) denote Data Fusion, Building Extraction, and Image-to-Image translation.

| Modality | Dataset | Year | Sensors | Resolution (m) | Types | Task | Domain Urban / Rural | Regions | Temporal |
|---|---|---|---|---|---|---|---|---|---|
| SAR | SEN12 (Schmitt et al., 2018) | 2018 | Sentinel-1 | 10 | VV | DF | Both | ≥ 80 | Dec 2016 – Nov 2017 |
| | SARptical (Wang & Zhu, 2018) | 2018 | TerraSAR-X | 1 | Unknown | DF | Urban | 1 | Jan 2009 – Dec 2013 |
| | SpaceNet6 (Shermeyer et al., 2020) | 2020 | Capella | 0.5 | HH, HV, VH, VV | BE | Urban | 1 | Aug 2019 |
| | MAVIC-T (Low et al., 2023a) | 2023 | GOTCHA | Unknown | Unknown | I2I | Unknown | 1 | Aug 2008 |
| | MAGIC (Low et al., 2024) | 2024 | GOTCHA, Umbra | Unknown | Unknown | I2I | Unknown | 4 | Unknown |
| | SAR2Earth (ours) | 2024 | Capella | 0.3 – 0.6 | HH | I2I | Both | 8 | Feb 2021 – Sep 2024 |
| EO | SEN12 (Schmitt et al., 2018) | 2018 | Sentinel-2 | 10 | RGB | DF | Both | ≥ 80 | Dec 2016 – Nov 2017 |
| | SARptical (Wang & Zhu, 2018) | 2018 | Aerial | 0.2 | RGB | DF | Urban | 1 | Jan 2009 – Dec 2013 |
| | SpaceNet6 (Shermeyer et al., 2020) | 2020 | WorldView-2 | 0.5 | P, MS, PS | BE | Urban | 1 | Aug 2019 |
| | MAVIC-T (Low et al., 2023a) | 2023 | Aerial | Unknown | RGB | I2I | Unknown | 1 | Aug 2008 |
| | MAGIC (Low et al., 2024) | 2024 | Aerial | Unknown | RGB | I2I | Unknown | 4 | Unknown |
| | SAR2Earth (ours) | 2024 | Google Earth | 0.15 – 0.6 | RGB | I2I | Both | 8 | Nov 2016 – Apr 2024 |

lack of diversity restricts the ability to generalize the performance of SAR-to-EO translation models across varying geographical contexts. Additionally, many of these datasets feature only one-day temporal differences between SAR and EO image pairs, which fails to reflect real-world data collection scenarios where temporal discrepancies can be substantial. Such discrepancies can arise from various factors, including satellite revisit intervals, cloud cover in EO imagery, and nighttime acquisition conditions.

As shown in Table 1, we present a summary of the characteristics of existing SAR and EO multi-modality datasets. These datasets are either not publicly available (Low et al., 2023a; 2024), limited in the number of regions they cover (Wang & Zhu, 2018; Shermeyer et al., 2020; Low et al., 2023a), have very low resolutions (Schmitt et al., 2018) that hinder generalization to objects like buildings, or do not consider real-world environments (Shermeyer et al., 2020; Low et al., 2023a; 2024).

To address these challenges, we introduce SAR2Earth, a comprehensive benchmark dataset for SAR-to-EO translation. SAR2Earth consists of spatially aligned SAR and EO images collected from 8 regions, encompassing both urban and rural environments. The dataset accounts for varying temporal differences between image pairs, reflecting realistic conditions encountered in practical applications. All codes and datasets are being made publicly available to support future research in this domain.

## 2 RELATED WORK

### 2.1 APPLICATIONS OF SAR IMAGERY

Numerous applications have been proposed to leverage SAR images across various domains. For instance, (Li et al., 2024) collected and labeled 100,000 SAR images to perform object detection directly on SAR data. Similarly, (Rambour et al., 2020) utilized spatially aligned SAR and EO images for multi-modal segmentation tasks, such as analyzing disasters like floods. Additionally, (Low et al., 2023b) focused on the classification of objects such as cars and buses within SAR imagery. Despite these efforts, SAR datasets face significant limitations. SAR data collection is costly and technically complex due to advanced radar technology, making SAR sensors more expensive and challenging than EO sensors. Processing SAR data is difficult due to speckle noise and other artifacts, requiring specialized expertise. These challenges hinder researchers and limit the creation of public datasets. Additionally, the lack of standardized datasets complicates widespread use, as SAR data varies in format and resolution depending on the provider, unlike standardized EO images.

### 2.2 CLOUD REMOVAL USING SAR IMAGERY

EO imagery cannot be used effectively when clouds are present, and to address this limitation, a cloud removal task using SAR data has been proposed. (Schmitt et al., 2019; Xu et al., 2023; Xia et al., 2024) introduced a benchmark dataset for cloud removal that uses multi-temporal EO images along with SAR imagery to remove clouds from EO data. However, this approach still cannot be used at night since EO imagery is unavailable during nighttime. Moreover, since SAR imagery is used as a condition or reference from an EO perspective, dynamic objects cannot be restored due to the time difference between SAR and EO acquisitions. To address these issues, the SAR-to-EO translation task is used, which aims to generate EO images using only the current SAR data.

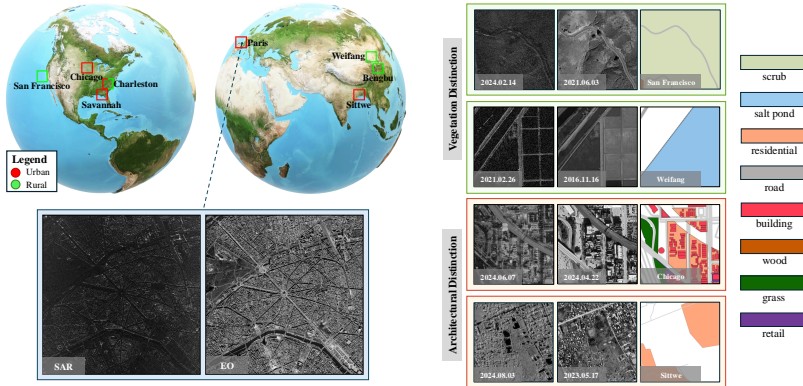

Figure 1: Geographic overview of the SAR2Earth dataset. This dataset highlights the diversity of geographic locations and environments, covering eight distinct regions—including Chicago, San Francisco, Charleston, Savannah, Paris, Bengbu, Weifang, and Sittwe—spanning both urban and rural areas across North America, Europe, and Asia. (As seen on the right, the consecutive columns represent SAR imagery, EO imagery, and OSM-based label maps.)

## 2.3 SAR-TO-EO TRANSLATIONS

To overcome the limitations of SAR datasets, SAR-to-EO translation techniques have been proposed. For instance, (Low et al., 2023a) introduced a method to utilize SAR images by translating them into EO images. To enhance the performance of SAR-to-EO translation, models such as Pix2Pix (Isola et al., 2017), Pix2PixHD (Wang et al., 2018), and CycleGAN (Zhu et al., 2017) have been employed. In applications such as Amazon deforestation monitoring (Cha et al., 2023), diffusion-based approaches (Rombach et al., 2022; Li et al., 2023) and generative adversarial networks (Isola et al., 2017; Wang et al., 2018) have been widely used for SAR-to-EO translation. Despite the numerous SAR-to-EO translation methods proposed, there has not been a rigorous comparison among paired methods, unpaired methods, and diffusion-based approaches. Furthermore, because the pre-processing and post-processing pipelines differ across studies, accurate analysis and benchmarking have been lacking.

## 2.4 REMOTE SENSING APPLICATIONS

Recent advancements in large foundation models and generalization models have brought significant benefits to satellite image analysis. GeoChat (Kuckreja et al., 2024) has demonstrated an EO (Electro-Optical) image-based language model by efficiently fine-tuning large language models. Segment Anything (Kirillov et al., 2023) introduced a segmentation model that can be utilized across any domain by training on billion-scale general vision datasets. These technologies have also been applied in the remote sensing domain, being used in various tasks such as change detection (Oh et al., 2023; Ding et al., 2024) and building segmentation (Osco et al., 2023). However, as revealed in the study (Yan et al., 2023), models based on Segment Anything and large language models like GeoChat do not perform effectively on SAR images due to their training on EO images, which have significantly different characteristics. Consequently, in the context of SAR imagery, the benefits of advancements in large foundation models and generalization models have not been fully harnessed.

## 3 SAR2EARTH DATASET

In this section, we provide a detailed description of the SAR2Earth dataset. The SAR2Earth dataset has the following key characteristics:

- **Global Data Collection for Generalization**: To evaluate generalization performance, the SAR2Earth dataset includes data collected from 8 regions across North America, Europe, and Asia.

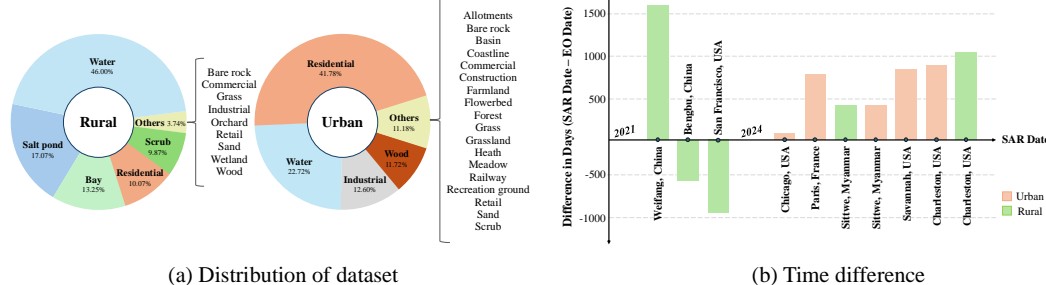

(a) Distribution of dataset          (b) Time difference

Figure 2: Statistics for the topological distribution and temporal differences in the dataset. (a) Distribution of urban and rural areas by topological elements. (b) Time differences between SAR and EO image captures across regions, indicating the satellite revisit cycles.

- **High Resolution Imagery**: The dataset consists of high resolution images, ranging from 0.15m to 0.6m, offering a diverse mix of spatial resolutions.

- **Consideration of Temporal Shifts**: The dataset accounts for a variety of temporal shifts, ranging from as close as a 1-month difference to as far as a 5-year gap, providing a wide spectrum of temporal scenarios.

- **Structural Diversity**: To address structural shifts, the data is divided into urban and rural categories. The classification is based on the ratio of buildings, amenities, and other structural elements, ensuring a balanced representation of diverse environments.

For sample images and detailed statistics of the dataset, please refer to Figure 1 and Figure 2.

## 3.1 DATASET DESIGN

**Data acquisition** SAR imagery is sourced from the Capella Space Open Data Program, with a resolution ranging from 0.3 to 0.6 meters per pixel. Its capability to capture detailed information irrespective of weather, cloud cover, or lighting makes it reliable for continuous monitoring.

EO imagery is obtained from Google Earth, with resolutions between 0.15 and 0.6 meters per pixel.

**SAR Pre-processing** SAR images require significant pre-processing to address noise (such as speckle), geometric distortions, and the wide dynamic range of pixel values. One of the critical steps is translating the raw amplitude or intensity values into decibels (dB), which enhances interpretability by compressing the dynamic range and providing a logarithmic representation suitable for further analysis. The conversion to decibels is performed using the following equation:

$$\sigma_{\text{dB}}^0 = 10 \log_{10}(S \cdot D^2) \tag{1}$$

where $\sigma_{\text{dB}}^0$ is the backscatter coefficient in decibels, $S$ is a scaling factor specific to the sensor, and $D$ is the calibrated digital number (DN) values in geocoded format. Note that $D$ is typically the square root of the intensity value, as SAR data is often represented in amplitude.

This conversion provides several benefits: it compresses the dynamic range for enhanced visualization, reduces the influence of extreme pixel values, and improves overall data interpretability, which are crucial for subsequent analysis steps.

**Co-registration of SAR and EO** A significant challenge in SAR-to-EO translation is achieving precise co-registration between the two image modalities due to inherent differences in both spatial resolution and coordinate systems. Accurate spatial alignment is essential to ensure that corresponding features in both modalities are matched correctly. To address the georeferencing discrepancies, both SAR and EO data are reprojected to a unified coordinate system, specifically the World Geodetic System 1984 (WGS84), which is the most widely adopted geodetic reference framework in remote sensing and geospatial applications. This reprojection guarantees spatial consistency, enabling accurate overlay and comprehensive analysis across both data types.

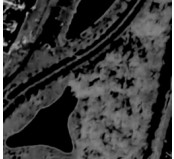 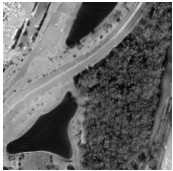 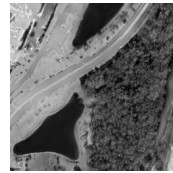 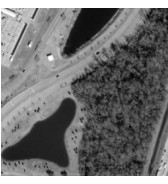

(a) SAR     (b) Denoised SAR     (c) Synthetic EO     (d) Refined EO     (e) EO

Figure 3: The results of SAR-to-EO translation at each step. (a) the original SAR image, (b) the denoised SAR, (c) the SAR-to-EO translation result, (d) the output from the refinement model, and (e) the EO image.

The co-registration process is performed using QGIS, a robust geographic information system platform. By leveraging the longitude and latitude coordinates inherent to WGS84, we executed image spatial alignment to achieve pixel-level precision. This procedure facilitated the precise synchronization of spatial features across SAR and EO imagery, thus enabling more effective translation and interpretation between the two data sources.

## 3.2 DATASET STATISTICS

To obtain detailed topological information, we utilized OpenStreetMap (OSM), classifying a total of 25 distinct land cover classes across all regions. The entire dataset covers a combined area of 1444.91 km$^2$. The dataset comprises a total of 99,998 images, each sized 256x256, generated with a stride of 128. For each region, the dataset is divided into training, validation, and test sets in a 7:1:2 ratio, as detailed further in the Appendix A. The regions are classified as urban if residential areas cover at least 25% of the total area. Additionally, if non-residential human-made areas, such as commercial, industrial, or retail spaces, occupy at least 5% of the total area, the region is also categorized as urban (Pesaresi et al., 2013; Esch et al., 2017; Wang et al., 2021).

As shown in Figure 2-(a), this classification provides an overview of the topological distribution of urban and rural areas. Specifically, rural areas predominantly consist of natural landscapes, such as vegetation and bodies of water, while urban areas are marked by the presence of human-made structures, including residential, commercial, and industrial buildings.

To assess the temporal diversity of our dataset, Figure 2-(b) illustrates the temporal differences between SAR and EO imagery acquisition across various regions. These temporal gaps vary significantly between regions, offering a wide range of temporal shifts. To the best of our knowledge, this makes our dataset the first to incorporate such diverse temporal differences across a broad set of geographic locations. Acquiring temporally aligned SAR-EO pairs without time discrepancies is particularly challenging in real-world settings, making this diversity crucial for practical applications.

## 4 SAR2EO PIPELINES

In this section, we provide a detailed explanation of our proposed SAR-to-EO pipeline. The SAR-to-EO baseline consists of three main stages: first, a de-noising step to remove the speckle noise inherent in SAR images, as shown in Figure 3-(b); second, an image-to-image translation module that translates SAR images into EO images, as illustrated in Figure 3-(c); and finally, a post-processing structure that refines the generated images for enhanced quality, as demonstrated in Figure 3-(d).

## 4.1 DE-NOISING

SAR images inherently contain speckle noise due to the interference of radar signals interacting with multiple scatterers. This noise has a multiplicative nature and is closely linked to the signal itself. Since speckle noise strongly correlates with neighboring pixels, conventional methods that assume noise and signal independence are less effective in removing it.

To address this, we adopt a blind-spot method, which predicts the clean value of a pixel based on its surrounding pixels rather than the noisy pixel itself. Given the high correlation of speckle

noise among neighboring pixels in SAR images, the blind-spot method is particularly effective at distinguishing and removing noise. This de-noising process enhances image quality for SAR-to-EO translation tasks.

In our work, we compare two blind-spot-based de-noising methods: (Lehtinen et al., 2018) and (Zhang et al., 2023).

### 4.2 IMAGE TO IMAGE TRANSLATION

SAR-to-EO image translation poses a complex challenge, requiring the handling of both paired and unpaired settings. Due to changes in ground conditions over time, achieving perfect temporal alignment between SAR and EO images is nearly impossible. For instance, while buildings and fixed structures remain relatively constant, elements like vegetation, moving objects, and lighting conditions vary, complicating precise registration.

Considering these factors, SAR-to-EO translation must effectively address both spatial alignment and temporal misalignment. In this paper, we compare paired and unpaired image-to-image translation approaches. Additionally, we propose a partially-paired image-to-image translation method by incorporating objective functions, such as MSE or MAE loss, into the unpaired setting. Given a SAR image $I_{sar}$ and an EO image $I_{eo}$, the modified loss function is defined as:

$$\mathcal{L}_{\text{total}}(G, D_{eo}, I_{sar}, I_{eo}) = \alpha\mathcal{L}_d(D_{eo}, I_{eo}, G(I_{sar})) + \beta\mathcal{L}_g(G, I_{sar}) + \gamma\mathcal{L}_{mse}(G(I_{sar}), I_{eo}) \quad (2)$$

Here, $\mathcal{L}_d$ is the discriminator loss, responsible for distinguishing real EO images $I_{eo}$ from generated EO images $G(I_{sar})$. The discriminator $D_{eo}$ learns this differentiation. $\mathcal{L}_g$ is the generator loss, applied to various unpaired image-to-image translation models such as CycleGAN (Zhu et al., 2017) and CUT (Park et al., 2020).

The term $\mathcal{L}_{mse}$ represents the MSE or MAE loss, which aims to minimize the reconstruction error between $G(I_{sar})$ and $I_{eo}$. By leveraging partially-paired data, this loss encourages the generator to produce EO images that closely resemble the real EO data, thereby reducing the differences between the generated and real images.

The terms $\alpha$, $\beta$ and $\gamma$ are all hyperparameters, and in all of our experiments, we set $\alpha$ and $\beta$ to 1, and $\gamma$ to 0.5.

### 4.3 POST-PROCESSING

After performing SAR-to-EO translation, the generated images may exhibit blurring or artifacts, especially when the data distribution differs from what is seen during training. However, models such as GeoChat or SAM often struggle to perform well on blurred or artifact-affected objects. Therefore, a refinement process is necessary to eliminate these artifacts.

We adopt Restormer as our refinement model. Let $D(.)$ represent the SAR-to-EO translation model, $G(.)$ the generator, and $R(.)$ the refinement network. The objective of the refinement step is defined as follows:

$$\mathcal{L}_{\text{refinement}} = \mathcal{L}_{\text{mae}}(R(G(D(I_{sar}))), I_{eo}) \quad (3)$$

## 5 EXPERIMENTS

In this section, we validate the SAR2Earth dataset using various image-to-image translation methods and experiment with different preprocessing and postprocessing techniques.

### 5.1 IMPLEMENTATION DETAILS

**Baselines** We selected Pix2Pix (Isola et al., 2017), Pix2PixHD (Wang et al., 2018), and the diffusion-based BBDM (Li et al., 2023) as paired baselines for image-to-image translation. Additionally, we chose CycleGAN (Zhu et al., 2017), CUT (Park et al., 2020), and StegoGAN (Wu et al., 2024) as unpaired baselines. All hyperparameters strictly followed the default settings of the

Table 2: Results for image-to-image translation baselines on the test set of SAR2Earth. We break down results by training data type: paired training data and unpaired training data. All models are trained on the train set of SAR2Earth.

| Model | Type | MAE ↓ | MSE ↓ | PSNR ↑ | SSIM ↑ | FID ↓ | LPIPS ↓ |
|---|---|---|---|---|---|---|---|
| Pix2Pix (Isola et al., 2017) | pair | 0.172 | 0.051 | 13.818 | 0.085 | 173.751 | 0.569 |
| Pix2PixHD (Wang et al., 2018) | pair | 0.151 | 0.041 | 15.319 | 0.162 | 155.073 | 0.564 |
| BBDM (Li et al., 2023) | pair | 0.161 | 0.047 | 14.772 | 0.163 | **123.051** | 0.477 |
| CycleGAN (Zhu et al., 2017) | unpair | 0.244 | 0.062 | 12.529 | 0.101 | 142.532 | 0.590 |
| CUT (Park et al., 2020) | unpair | 0.236 | 0.086 | 11.172 | 0.094 | 144.312 | 0.592 |
| StegoGAN (Wu et al., 2024) | unpair | 0.214 | 0.073 | 12.041 | 0.152 | 158.930 | 0.595 |
| CycleGAN (Zhu et al., 2017) | pair+unpair | 0.189 | 0.063 | 13.592 | 0.109 | 142.532 | 0.540 |
| CUT (Park et al., 2020) | pair+unpair | **0.132** | **0.039** | **16.500** | **0.199** | 140.227 | **0.350** |
| StegoGAN (Wu et al., 2024) | pair+unpair | 0.197 | 0.059 | 14.213 | 0.161 | 166.325 | 0.593 |

respective methods [1][2][3][4]. We refer to the output of SAR-to-EO models as *SynEO*, and the approach combining paired and unpaired methods is termed the *hybrid* method.

**Experiments settings** Table 2 presents results obtained without applying de-noising or post-processing, providing a baseline for comparison. From Table 3 onward, de-noising and post-processing steps are consistently applied, utilizing Hybrid CUT to enhance model performance. This progression demonstrates the impact of these additional steps, ensuring clarity in the experimental setup and the effects of de-noising and post-processing on SAR-to-EO translation performance.

We use the official codes for OpenEarthMap (Xia et al., 2023) and GeoChat, where the Unet-Former (Wang et al., 2022b) model are used for land cover segmentation, and the 7B model are used for GeoChat. For further details on the experimental setup of land cover segmentation, please refer to Appendix C. We strictly followed all the hyperparameters and settings from the original code.

**Evaluation metrics** To evaluate the performance of the SAR-to-EO image translation task, we use MAE (Mean Absolute Error), MSE (Mean Squared Error), PSNR (Peak Signal-to-Noise Ratio), and SSIM (Structural Similarity Index Measure) to measure pixel-level accuracy and structural similarity. These metrics capture the absolute and squared differences between the generated and real EO images, assess image quality in terms of noise (PSNR), and ensure structural consistency (SSIM), which are crucial for maintaining fidelity in pixel values and structures in SAR-to-EO translation.

Additionally, we use FID (Fréchet Inception Distance) and LPIPS (Learned Perceptual Image Patch Similarity) to evaluate the perceptual quality and realism of the generated EO images. FID assesses the similarity in feature distributions between the generated and real EO images, while LPIPS focuses on perceptual differences based on deep feature representations, ensuring that the generated images visually resemble real EO data.

## 5.2 COMPARISON OF BASELINE

Table 2 presents the results of comparing image-to-image translation methods on the SAR2Earth dataset. As observed in the comparison table, methods under the *paired* setting achieved high accuracy results (MSE, MAE). In contrast, methods under the *unpaired* setting showed lower accuracy (MSE, MAE) but attained higher perceptual scores (FID).

The SAR2Earth task aims to accurately *predict* the correct EO image rather than simply *generate* plausible images. Therefore, metrics such as perceptual scores and MSE, MAE are both important. Accordingly, we combined unpaired baselines that achieved high perceptual scores with paired methods that obtained high MSE and MAE performance. We conducted experiments by applying Eq. 2 on the paired images using existing unpaired methods such as CycleGAN, CUT, and StegoGAN.

---

[1]https://github.com/junyanz/pytorch-CycleGAN-and-pix2pix

[2]https://github.com/taesungp/contrastive-unpaired-translation

[3]https://github.com/xuekt98/BBDM

[4]https://github.com/sian-wusidi/StegoGAN

Experimental results showed that the hybrid CUT in Table 2 achieved the highest performance. This is because the SAR2Earth dataset is spatially aligned but temporally unaligned. As a result, objects like buildings are in a paired setting, while moving objects are in an unpaired setting. Therefore, a baseline that considers both settings achieved the best performance.

## 5.3 COMPARISON OF PROCESSING

Table 3: Ablation study on de-noising preprocessing methods.

| Model | De-noising | MAE ↓ | MSE ↓ | PSNR ↑ | SSIM ↑ | FID ↓ | LPIPS ↓ |
|---|---|---|---|---|---|---|---|
| | MedianBlur | 0.122 | 0.037 | 16.907 | 0.219 | 140.530 | 0.342 |
| CUT | GaussianBlur | 0.126 | 0.032 | 16.526 | 0.222 | 140.172 | 0.348 |
| (pair+unpair) | Noise2Noise (Lehtinen et al., 2018) | 0.114 | 0.029 | 16.683 | 0.225 | 144.230 | 0.344 |
| | MM-BSN (Zhang et al., 2023) | **0.107** | **0.022** | **17.431** | **0.236** | **136.684** | **0.332** |

**Comparison of de-noising**   SAR images contain a large amount of speckle noise. This noise appears as granular interference, obscuring important features and textures in the image. It complicates the feature extraction process in data-driven models by introducing high-frequency artifacts, making it challenging to learn accurate mappings between SAR and EO images. To address this issue, de-noising methods have been applied, but because elements in SAR images that appear as noise can actually be important signals, de-noising methods need to be applied carefully. Table 3 shows the performance variations of SAR-to-EO translation according to different de-noising methods.

The results in Table 3 demonstrate that as the de-noising methods become more advanced, performance improves. These experimental results indicate that in the SAR-to-EO translation task, employing more advanced de-noising methods positively impacts performance.

**Comparison of refinement**   We compared the performance of SAR-to-EO translation with respect to post-processing. For post-processing, we used (Zamir et al., 2022), and during training, we aimed for refinement by adding random deformations (affine transforms, random Gaussian noise) to the EO images. After that, we applied a refinement model to the images translated from SAR-to-EO. We observed that the FID score decreased from **136** to **128**, indicating an improvement in perceptual quality, while the other scores did not change significantly. As observed in the results, we confirmed that the performance improved slightly. Figure 3 illustrates (a) the original SAR, (b) the denoised SAR, (c) the synthetic EO, (d) the refined EO, and (e) the ground truth EO. As shown in Figure 3, we confirmed that the artifacts present in (c) disappeared in (d) through refinement. These experimental results indicate the cause of the performance improvement due to refinement.

## 5.4 MODEL GENERALIZATION EVALUATION

The characteristics of SAR images vary significantly by region due to radar backscatter, making it difficult to distinguish between surfaces with similar structures, like oceans and flat areas. As a result, domain gaps in SAR data are often larger than in EO imagery. To evaluate this, we conduct in-domain experiments by training and testing models within the same region.

Urban areas, with their complex structures, present larger domain gaps compared to rural areas, which tend to have more uniform natural features. As shown in Table 4, rural regions generally outperform urban areas in in-domain evaluations across all metrics. Notably, training on combined urban regions often yields better results than training on a single region, likely due to increased data diversity. However, for rural regions, training on individual regions produces better results, suggesting that localized models perform better for natural features.

In cross-domain experiments (Urban → Rural and Rural → Urban), we observe significant performance drops, emphasizing the large differences between these domains. Thus, for practical applications, collecting and training data tailored to specific regional characteristics is more beneficial than simply expanding the dataset without considering regional uniqueness.

## 5.5 QUALITATIVE RESULTS

Figure 4 qualitatively compares the results of SAR-to-EO translation across different baselines. As shown in the figure, CUT (hybrid) produces the most visually plausible results. Specifically, in the

Table 4: Results for regional test set when trained with 10 regions or the entire urban (Charleston-U, Chicago, Paris, Savannah, Sittwe-U) and rural regions (Bengbu, Charleston-R, San Francisco, Sittwe-R, Weifang).

| Experiment Setting | Region | MAE ↓ | MSE ↓ | PSNR ↑ | SSIM ↑ | FID ↓ | LPIPS ↓ |
|---|---|---|---|---|---|---|---|
| In-Domain (Single region) | Charleston-U | 0.108 | 0.030 | 17.235 | 0.230 | 130.582 | 0.320 |
| | Chicago | 0.112 | 0.033 | 16.983 | 0.225 | 132.467 | 0.327 |
| | Paris | **0.105** | **0.029** | **17.301** | **0.235** | **128.430** | **0.315** |
| | Savannah | 0.115 | 0.034 | 16.875 | 0.222 | 135.098 | 0.330 |
| | Sittwe-U | 0.109 | 0.031 | 17.102 | 0.229 | 131.744 | 0.322 |
| | Bengbu | 0.098 | 0.025 | 18.512 | 0.240 | 120.320 | 0.300 |
| | Charleston-R | 0.101 | 0.027 | 18.301 | 0.238 | 123.982 | 0.308 |
| | San Francisco | 0.097 | 0.024 | 18.734 | 0.242 | 118.567 | 0.295 |
| | Sittwe-R | 0.099 | 0.026 | 18.589 | 0.239 | 121.765 | 0.305 |
| | Weifang | **0.096** | **0.023** | **18.852** | **0.245** | **117.231** | **0.292** |
| In-Domain | Urban→Urban | 0.106 | 0.028 | 17.478 | 0.240 | 125.345 | 0.310 |
| | Rural→Rural | 0.097 | 0.024 | 18.715 | 0.241 | 115.984 | 0.298 |
| Cross-Domain | Urban→Rural | 0.135 | 0.043 | 16.253 | 0.210 | 145.450 | 0.360 |
| | Rural→Urban | 0.132 | 0.041 | 16.438 | 0.218 | 143.890 | 0.355 |

Figure 4: Qualitative comparison of various image-to-image translation methods for SAR-to-EO translation in rural and urban cases.

second row, indicated by the green dotted box, the SAR image does not contain an airplane signal, and all baselines succeed to generate an airplane in their corresponding SAR-to-EO translation outputs. This experiment demonstrates that, despite the temporally unaligned nature of the SAR-to-EO setting, combining paired and unpaired training approaches effectively mitigates this challenge.

In the rural example (third row), all baselines produce more plausible images compared to their urban counterparts. However, as highlighted by the red dotted line, fully paired methods like pix2pix and pix2pixHD tend to distort features. This is due to the differing imaging angles between SAR and EO data, where SAR images are often captured from a perspective distinct from that of EO imagery. As a result, the paired models attempt to generate EO-like angles, even for features not present in the original SAR image, creating non-existent structures in the SynEO output. In contrast, baselines that combine paired and unpaired approaches do not exhibit this distortion tendency, maintaining consistency with the original SAR imagery. These results suggest that if the goal is to generate EO-like angles from SAR data, a paired setting is optimal. However, if the aim is to faithfully replicate the appearance of SAR imagery, a combined paired and unpaired training approach is more effective.

## 5.6 APPLICATION

**GeoChat** Figure 5 illustrates the results of testing SAR images, synthetic EO (SynEO) images obtained through SAR-to-EO translation, and actual EO images using the GeoChat large language model (LLM). As shown in the figure, when a SAR image is input into GeoChat, the responses from the model contain entirely incorrect content. This indicates a failure to interpret the SAR data accurately, primarily because SAR images are excessively noisy and differ significantly from the EO or RGB images on which LLMs are predominantly trained. In contrast, when the SynEO and

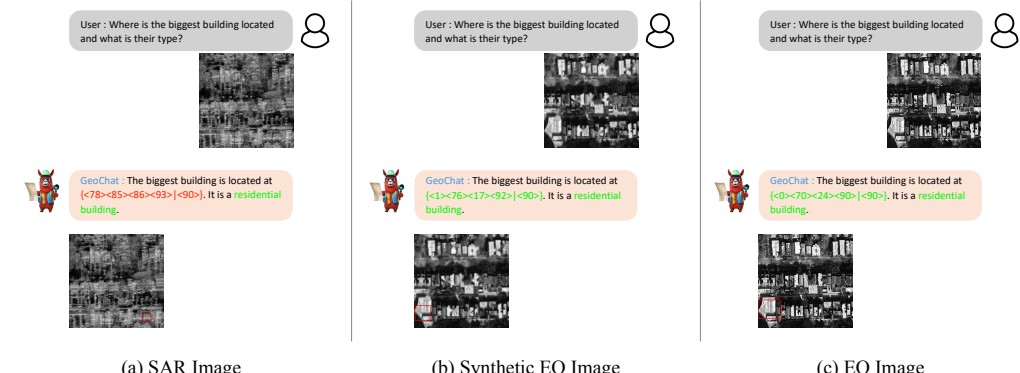

(a) SAR Image          (b) Synthetic EO Image          (c) EO Image

Figure 5: Comparison of visual grounding tasks using SAR, EO, and SynEO. (a) is the SAR image, (b) is SynEO, and (c) is EO.

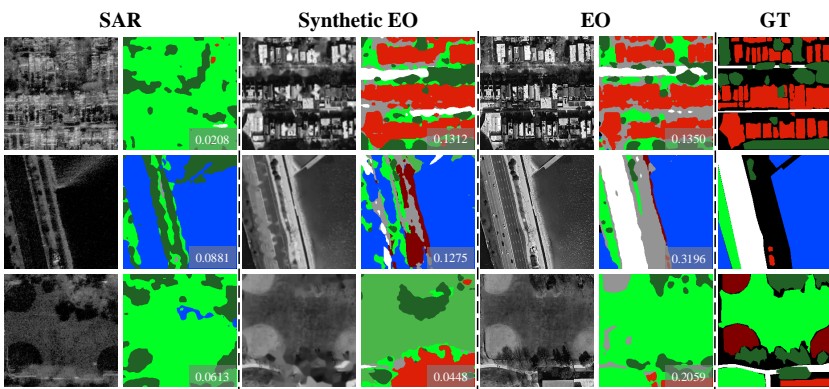

Figure 6: Inference results of SAR, SynEO, and EO images using UnerFormer trained on grayscale OpenEarthMap. The bottom-right corner of each prediction shows the mIoU score.

EO images are provided as input, GeoChat generates correct answers, demonstrating its ability to understand and analyze these images effectively.

**Land Cover Segmentation** As shown in Figure 6, the results of land cover segmentation demonstrate that using synthetic EO (SynEO) images achieves higher accuracy than SAR images for classes such as buildings or roads. However, as observed in the third row, SAR images outperform SynEO in classes related to natural landscapes, such as grass. We hypothesize that this is because models trained on grayscale images, when applied to SAR images, interpret the textures in SAR images as being similar to natural elements like trees or grass.

## 6 CONCLUSION

In this paper, we present SAR2Earth, a novel public benchmark dataset for Synthetic Aperture Radar to Electro-Optical (SAR-to-EO) translation, aiming to support a wide range of remote sensing applications. We systematically evaluate SAR2Earth by applying various state-of-the-art image-to-image translation models and provide comprehensive benchmark results. Furthermore, we conduct extensive ablation studies—from SAR data pre-processing to model architecture design—to offer valuable insights into the effective utilization of SAR data. Additionally, we validate the versatility of SAR2Earth through experiments with GeoChat and SegmentAnything, demonstrating the potential of SAR-to-EO translation in enhancing data accessibility and utility. Finally, we publicly release our dataset and code to facilitate and encourage future research in this domain. We hope that our research will be widely utilized in tasks such as disaster response and AI for social good.

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

# A    DATASET DETAILS

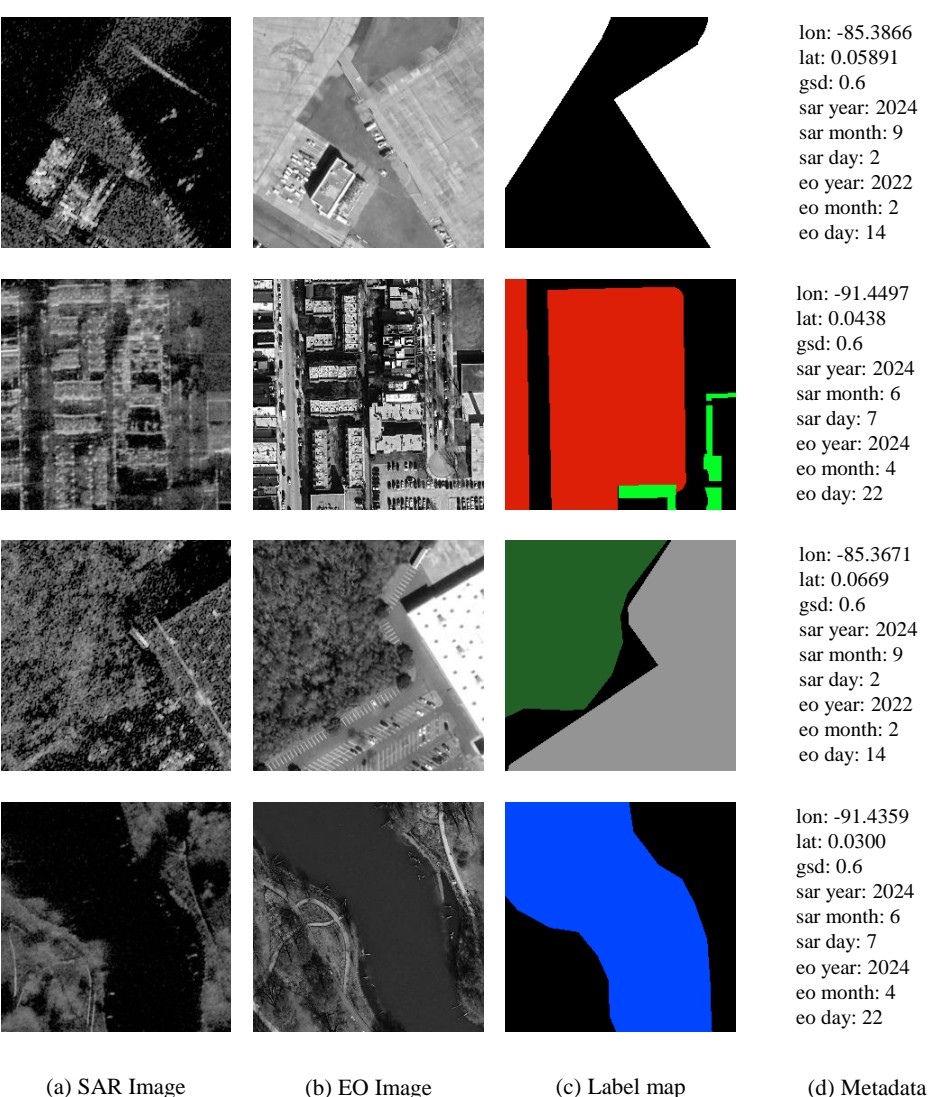

|          |          |             |               |
|----------|----------|-------------|---------------|
| (a) SAR Image | (b) EO Image | (c) Label map | (d) Metadata |

Figure 7: Overview of the SAR2Earth dataset components.

## A.1    DATA SPLITS

The SAR2Earth dataset is divided into train, validation, and test sets based on spatial regions to ensure unbiased evaluation and consistency. The splits are defined as 70% for training, 10% for validation, and 20% for testing, considering the spatial distribution of each region to ensure robust generalization. This structured splitting approach ensures robust performance evaluation in SAR-to-EO translation tasks.

## A.2  Label maps and metadata

The label maps and metadata in the SAR2Earth dataset offer vital information for interpreting and analyzing the data. The label maps represent topological information, categorizing land cover classes derived from OpenStreetMap (OSM) data. Each label map is spatially aligned with its corresponding SAR and EO image pair, facilitating comprehensive spatial and semantic analysis.

The metadata includes fields such as geographic coordinates (based on the image center), ground sampling distance (GSD), and acquisition dates for both SAR and EO images. These components provide detailed context for each image pair, supporting diverse remote sensing applications. Figure 7 presents examples of dataset components, including label maps and metadata, for better understanding.

# B  Task Overview and Pipeline

## B.1  Task definition

Table 5 presents a taxonomy of tasks utilizing SAR. As shown in the table, image-to-image translation (I2I) and cloud removal (CR) produce the same output, $I_{SynEO}$, but differ in their inputs and the point in time at which inference is made. For instance, I2I takes SAR as input to generate the corresponding synthetic EO (SynEO), while CR typically uses past SAR data along with multi-temporal EO and cloudy EO inputs to generate a CleanEO output. Thus, the inference point of time for CR aligns with EO. In contrast, for all other tasks, the inference point of time aligns with SAR.

Table 5: Taxonomy of tasks utilizing SAR

| Task | Input | Multi-tmporal | Models | Output | Inference point of time |
|------|-------|---------------|--------|--------|-------------------------|
| I2I | $I_{SAR}$ | - | Generator | $I_{SynEO}$ | SAR |
| CR | $I_{SAR}$, $I_{EO}$ | ✓ | Generator | $I_{SynEO}$ | EO |
| OD | $I_{SAR}$ | - | Detector | BBOX | SAR |
| SEG | $I_{SAR}$ | - | Segmenter | MASK | SAR |

## B.2  Flow Chart

Figure 8 shows the pipeline of our SAR-to-Earth approach along with image samples from each step. As depicted in the figure, all our models are connected sequentially, and during actual inference, only SAR images are used.

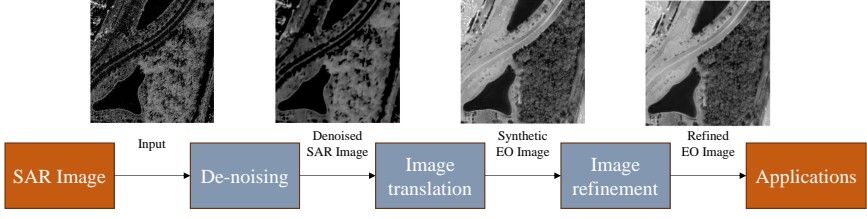

Figure 8: Overview of the SAR-to-EO translation pipeline.

# C  Details of Land Cover Segmentation

We utilized the official code [5] and dataset provided by OpenEarthMap (OEM). The model was trained on the entire dataset, including the xView2 Dataset, while adhering to the data splits defined

---

[5] https://github.com/bao18/open_earth_map

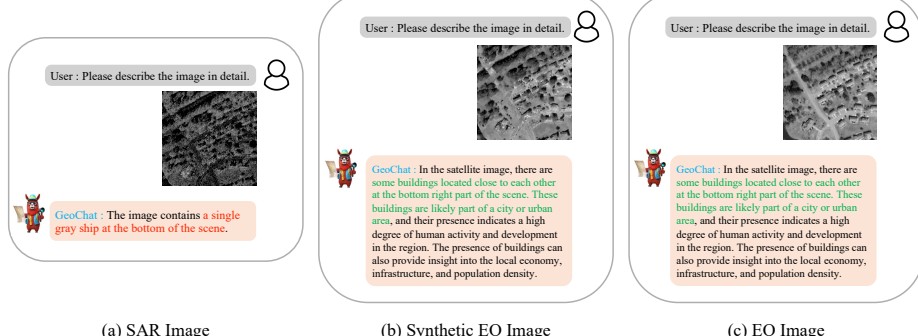

Figure 9: Interpretation differences between SAR and EO images. The SAR image (a) leads to an incorrect identification of objects in the scene, such as mislabeling a ship, while the synthetic EO (b) and real EO (c) images accurately capture key urban features, including clusters of buildings.

in the official code. To adapt the training process for our target dataset, the images were resized to 256×256 and converted to grayscale before training.

## C.1 CLASS MAPPING

Due to the differences between the labels in our dataset and those provided by the OEM dataset used for training, we performed a class mapping process. The detailed mapping scheme is as follows:

- **Cropland**: Includes *allotments*, *farmland*, *greenhouse_horticulture*, *orchard*, *plant_nursery*, and *vineyard*.
- **Pavement**: Includes *apron*, and *runway*.
- **Bareland**: Includes *bare*, *bare_rock*, and *sand*.
- **Water**: Includes *bay*, *salt_pond*, and *water*.
- **Buildings**: Includes *construction*, *commercial*, *industrial*, *residential*, *retail* and *farmyard*.
- **Grass**: Includes *flowerbed*, *grass*, *grassland*, *heath*, *meadow*, *national_park*, and *scrub*.
- **Tree**: Includes *forest* and *wood*.

## D ADDITIONAL QUALITATIVE RESULTS

This section presents additional qualitative results that can not be included in the main manuscript.

## D.1 IMAGE INTERPRETATION

Figure 9 presents the results of image interpretation. As shown in the figure, GeoChat, trained on large-scale EO images, failed to provide accurate results for SAR images but generated correct responses for SynEO images. These findings indicate that SynEO can enhance the applicability of models trained on large-scale EO datasets, bridging the gap between SAR and EO data.

## D.2 SEGMENT ANYTHING

We tested the Segment Anything model (Kirillov et al., 2023) on SAR, SynEO, and EO images. As shown in Figure 10, the model struggled to perform effectively on SAR images due to the presence of speckle noise and irregular patterns, which significantly hindered segmentation performance. In contrast, the SynEO images led to successful segmentation by the Segment Anything model. Although the results were still not on par with the EO images, the performance showed substantial qualitative improvement. Improving the SAR-to-EO translation quality could enable generalized models like Segment Anything to be more effectively utilized on SAR data.

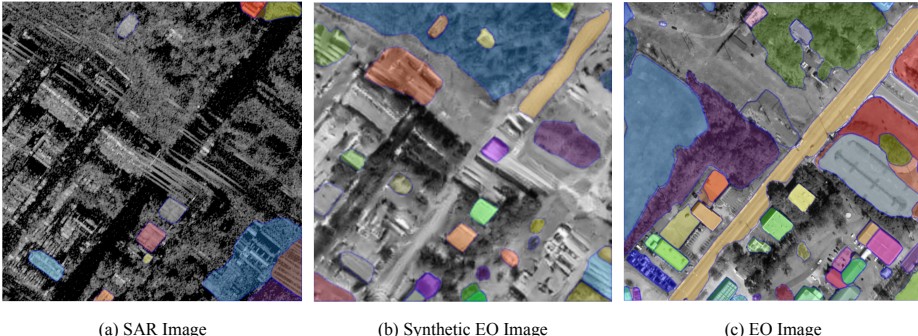

(a) SAR Image          (b) Synthetic EO Image          (c) EO Image

Figure 10: Segmentation results using SAM (Segment Anything Model) for different modalities.

# E  LIMITATIONS AND BORDER IMPACT

## E.1  LIMITATIONS AND FUTURE WORKS

While SAR images may appear visually similar across regions, their actual representations can differ significantly due to variations in surface roughness. Therefore, regional characteristics play a critical role in SAR-to-EO translation. However, in this work, we do not leverage extra modalities, such as OSM-based label maps or metadata (longitude, latitude, ground sample distance, and date), which can be used to better account for these regional differences.

The SAR2Earth dataset does provide metadata information for each image pixel level, allowing for future research to utilize this additional data. As part of our future work, we plan to incorporate extra modality-based regional features to enhance the performance of SAR-to-EO translation models by making them more sensitive to regional variations.

## E.2  BORDER IMPACT

Our work holds significant implications for remote sensing and related fields. By providing a public benchmark dataset, we aim to accelerate research in SAR-to-EO translation, facilitating advancements in environmental monitoring, disaster response, and urban development. SAR2Earth is particularly valuable in disaster scenarios like floods, where heavy cloud cover renders traditional EO imagery less effective. Since SAR can penetrate clouds and is unaffected by weather conditions or daylight, translating SAR-to-EO images can provide critical information when it is most needed. By translating SAR images into EO-like images, we facilitate the application of advanced AI models developed for EO imagery to SAR data, potentially maximizing the utility of established methodologies.

