# OpenReview forum: "SAR2Earth: A SAR-to-EO Translation Dataset for Remote Sensing Applications"
_ICLR.cc/2025/Conference — Submitted to ICLR 2025_

### Official Review · Reviewer_LqKw · 2024-10-16

**Soundness:** 2
**Presentation:** 3
**Contribution:** 1
**Rating:** 5
**Confidence:** 3

**Summary:**

Electro-optical (EO) satellite images are used in many important applications of machine learning. But the Earth's surface can be blocked by cloud coverage and is not viewable at night. Synthetic aperture radar (SAR) do not have these two limitations. This paper introduces SAR2Earth, a dataset for SAR-to-EO image translation. It consists of 18 spatially registered pairs of SAR and EO images from 8 different regions. This paper also benchmarks image-to-image translation methods (introduced for natural imagery) on SAR2Earth, showing that the CUT method performs best. This paper qualitatively shows that EO foundation model predictions improve by translating a SAR image to an EO image, then providing the foundation model with this synthetic EO image — compared to directly providing the SAR image.

**Strengths:**

The paper is well written and presented. Remote sensing is an important application area of ML. The introduced dataset — named SAR2Earth — is valuable to the remote sensing community, as it contains high-resolution SAR-EO pairs sampled from diverse regions and temporal differences. Running many baseline methods on SAR2Earth is a valuable contribution. The examples using SAR to EO translation to create synthetic EO images which are fed to foundation models is also a valuable contribution — and it shows that this research direction complements broader trends in ML.

**Weaknesses:**

My main concern is that I do not believe this paper is valuable to share with the broader ICLR community. I believe it is more suited to RS-specific journals or workshops. The most similar work to this, that I am ware of, is a CVPR 2024 workshop paper by Low et al., that is cited. SAR2Earth seems to have more regional diversity and this paper performs more extensive (and valuable) experiments. As far as I can tell, Low et al. provide far more SAR-EO pairs than this submission.

Other comments:
- It is hard to tell what the MAE / MSE/ PSNR metrics imply for a remote sensing practitioner. For example, are synthetic images with an MAE of 0.1 still useful? One potential experiment is to find a dataset with SAR-EO pairs, like BigEarthNet or EuroSat (there is a SAR version available). Then translate the SAR image to EO and try classifying it. The performance drops would be very interesting to me.
- I cannot find a discussion on RS models that were pretrained to ingest SAR. There are many: Presto, CROMA, DOFA, SoftCon, DeCur, etc. If we have foundation models that can ingest SAR, wouldn't it be better to directly provide the SAR image, rather than a synthetic EO image that requires a SAR-to-EO translator?

**Questions:**

It seems like this dataset only contains 18 SAR-EO pairs, can you confirm this is the case? If so, what is the bottleneck to scale your collection pipeline up to many thousands of pairs?

Please also see questions in the weaknesses section.

**Details Of Ethics Concerns:**

I have no concerns.

---

> ### Author Response · Authors · 2024-11-21
> **Response to reviewer LqKw (1)**
>
> We sincerely thank the reviewer for their thoughtful feedback and recognition of the strengths of our work. We are especially grateful for acknowledging the importance of remote sensing as an application area of machine learning and the value of the SAR2Earth dataset to the community. Your appreciation of our high-resolution SAR-EO pairs, diverse sampling, and temporal differences encourages us greatly. Additionally, we are pleased that you found our baseline evaluations and the use of synthetic EO images to enhance foundation model predictions to be valuable contributions. Your feedback motivates us to continue advancing this research direction.
>
>
> > Weakness 1 : My main concern is that I do not believe this paper is valuable to share with the broader ICLR community. I believe it is more suited to RS-specific journals or workshops. The most similar work to this, that I am ware of, is a CVPR 2024 workshop paper by Low et al., that is cited. SAR2Earth seems to have more regional diversity and this paper performs more extensive (and valuable) experiments. As far as I can tell, Low et al. provide far more SAR-EO pairs than this submission.
>
> While **Low et al.'s datasets, MAVIC-T and MAGIC (as referenced in our Table 1)**, contribute valuable resources, they are more limited in several aspects compared to SAR2Earth. MAVIC-T contains data from only **one region**, and MAGIC includes **four regions**, which is significantly fewer than the 18 regions covered by SAR2Earth. Moreover, the data coverage of MAGIC is, on average, **24.2 times smaller** than that of SAR2Earth.
>
> Another key distinction is that SAR2Earth provides clear separations between **rural and urban areas**, offering a broader diversity of scenes and enhancing its applicability for various tasks. In contrast, the MAVIC datasets are confined to a few specific regions, which may limit their generalizability. We believe these differences highlight the broader scope and utility of SAR2Earth for the community.
>
> > Weakness 2 : It is hard to tell what the MAE / MSE/ PSNR metrics imply for a remote sensing practitioner. For example, are synthetic images with an MAE of 0.1 still useful? One potential experiment is to find a dataset with SAR-EO pairs, like BigEarthNet or EuroSat (there is a SAR version available). Then translate the SAR image to EO and try classifying it. The performance drops would be very interesting to me.
>
> We understand the concern about the lack of intuitive interpretability of metrics like MAE, MSE, and PSNR in the remote sensing domain. To provide more meaningful insights, we evaluated the performance of SAR-to-EO translation using a Land Cover Segmentation task, as shown in **Figure 6**.
>
> Our results indicate that segmentation mIoU improves when using synthetic EO images generated through SAR-to-EO translation, compared to using SAR images directly. However, the performance remains lower than that achieved with original EO images. These findings highlight the practical value of SAR-to-EO translation while acknowledging the room for further improvement.

---

> ### Author Response · Authors · 2024-11-21
> **Response to reviewer LqKw (2)**
>
> > Weakness 3 : I cannot find a discussion on RS models that were pretrained to ingest SAR. There are many: Presto, CROMA, DOFA, SoftCon, DeCur, etc. If we have foundation models that can ingest SAR, wouldn't it be better to directly provide the SAR image, rather than a synthetic EO image that requires a SAR-to-EO translator?
>
> While directly utilizing SAR data through foundation models like Presto, CROMA, or DOFA is a valuable option, there are several reasons why SAR-to-EO translation remains advantageous:
>
> 1. Interpretability Challenges: SAR data is inherently complex and difficult for non-experts to interpret. SAR-to-EO translation bridges this gap by converting SAR into a more familiar and accessible EO-like format, simplifying downstream analysis.
>
> 2. Limited SAR Resources: Unlike EO imagery, which benefits from extensive datasets, pre-trained models, and open-source tools, SAR datasets are scarce, and SAR-specific foundation models, such as EarthGPT, often lack open-source implementations or pre-trained weights, limiting their accessibility.
>
> 3. Handling SAR Polarizations: Foundation models like Presto, CROMA, and DOFA do not universally handle all SAR polarizations (e.g., HH, VV, VH) in a single framework. SAR-to-EO translation bypasses these complexities, enabling the use of established EO analysis pipelines.
>
> 4. Accessibility and Usability: EO imagery is supported by a vast ecosystem of tools and pre-trained models, making it far more accessible for various tasks. SAR-to-EO translation allows practitioners to leverage this ecosystem without needing extensive expertise in SAR-specific data.
>
> 5. Fine-Tuning Requirements: Even with foundation models, fine-tuning for specific applications is necessary, which requires suitable datasets. SAR-to-EO translation facilitates this process by transforming SAR data into a format compatible with existing EO-focused datasets and methods, streamlining model development.
>
> While SAR-specific foundation models are a promising direction, SAR-to-EO translation complements these efforts by broadening accessibility, reducing complexity, and enabling a seamless integration of SAR data into established EO workflows.
>
> > Question 1: It seems like this dataset only contains 18 SAR-EO pairs, can you confirm this is the case? If so, what is the bottleneck to scale your collection pipeline up to many thousands of pairs?
>
> Our SAR2Earth dataset contains a total of 99,998 SAR-EO pairs, divided into 69,989 for training, 20,018 for testing, and 9,991 for validation. The reference to "18 pairs" in the text refers to the number of original imagery, each of which is very large in size. When these are divided into 256×256 patches, the total number of pairs scales up to approximately 100,000.
>
> This scalable data collection method ensures a large and diverse dataset, which is critical for effective and authentic SAR-to-EO image generation. We have clarified this point in the main text **(Section 3.2)** and provided additional details in **Appendix A** to avoid any misunderstandings.

---

> > ### Comment · Reviewer_LqKw · 2024-11-23
> >
> > I thank the authors for their thoughtful rebuttal. In particular, it clarifies the dataset size. Indeed, 100 K high-resolution SAR-RGB pairs are significantly valuable to the RS community. And beyond image translation, this dataset could be used in a larger corpus of unlabelled RS data to pretrain RS models.
> >
> > However, I am not convinced this contribution is worth sharing with the broader ICLR community. To the best of my knowledge, this submission is more suited to a CVPR workshop, the NeurIPS dataset track, or RS-specific venues.
> >
> > I will raise my score from 3 to 5.

---

> > > ### Author Response · Authors · 2024-11-29
> > > **Response to reviewer LqKw**
> > >
> > > Thank you for understanding us and helping improve our score!!  We will take full responsibility for our work and manage the code and data thoroughly until the end.

---

### Official Review · Reviewer_bjda · 2024-10-20

**Soundness:** 3
**Presentation:** 2
**Contribution:** 3
**Rating:** 5
**Confidence:** 4

**Summary:**

This paper introduces SAR2Earth, a benchmark dataset designed for the translation of synthetic aperture radar (SAR) images into electro-optical (EO) image representations. This dataset addresses the limitations of EO images, which cannot penetrate cloud cover or capture nighttime imagery, by leveraging the all-weather and day-night capabilities of SAR images.

**Strengths:**

1. This article has strong practical applications and provides a broad platform for future research.
2. Their dataset have considered realistic conditions for practical applications.
3. This paper have provided extensive ablation studies from pre-processing to post-processing techniques.

**Weaknesses:**

1. As a paper introducing a new dataset, this article provides too few qualitative results.
2. The number of images in the proposed dataset remains unknown, and details such as image numbers, the size of the training and test set are not disclosed.
3. The applications demonstrated in this paper are limited in section 5.6.

**Questions:**

1. How to ensure the accuracy of the co-registration of SAR and EO?
2. Why not compared diffusion-based approaches for SAR-to-EO translation? These methods are widely used (Line 134).
3. For SAR-to-EO pipeline proposed in this paper, how can the authenticity of the synthetic EO images be ensured, considering that generative models may introduce artifacts? If the model generates false features, how should they be handled?

---

> ### Author Response · Authors · 2024-11-21
> **Response to reviewer bjda**
>
> We sincerely thank the reviewer for highlighting the strengths of our work. We deeply appreciate the recognition of the practical applications of the SAR2Earth dataset and its potential to serve as a broad platform for future research. We are also grateful for the acknowledgment of our efforts to consider realistic conditions and for providing extensive ablation studies covering both pre-processing and post-processing techniques. reviewer's feedback encourages us to continue refining and improving our work.
>
> > Weakness 1 : As a paper introducing a new dataset, this article provides too few qualitative results.
>
> > Weakness 2 : The number of images in the proposed dataset remains unknown, and details such as image numbers, the size of the training and test set are not disclosed.
>
> > Weakness 3 : The applications demonstrated in this paper are limited in section 5.6.
>
> We have added qualitative results for the Visual Grounding task and Land Cover Segmentation to the main manuscript, while moving the existing qualitative results to the supplementary material. Additionally, we have included more qualitative examples in **Figure 4** and provided detailed information about the dataset, including the number of images and data splits, in **Section 3.2** and **Appendix A**. We hope these updates address your concerns and provide greater clarity.
>
> > Question 1: How to ensure the accuracy of the co-registration of SAR and EO?
>
> To ensure accurate co-registration between SAR and EO images, we align their coordinate systems and use longitude and latitude information for co-registration. This straightforward method ensures consistency across the entire dataset.
>
> However, as resolution increases, co-registration accuracy can be affected by factors like satellite imaging angles, which may introduce misalignments. While feature matching-based methods could address these challenges, they often require additional training and may perform inconsistently depending on the scene.
>
> Since the primary focus of our work is on dataset creation, we opted for a method-independent, pixel-matching strategy based on geographic coordinates. This approach ensures consistent and reliable performance across diverse regions and is practical for real-world SAR-to-EO applications.
>
> > Question 2 : Why not compared diffusion-based approaches for SAR-to-EO translation? These methods are widely used (Line 134).
>
> In our experiments, we included the BBDM [1] model, a diffusion-based approach that represents the state-of-the-art in image-to-image translation. This model is widely utilized in SAR-to-EO translation tasks, as demonstrated in recent works such as MultiEarth2023 [2] and Flood [3].
>
> >  Question 3: For SAR-to-EO pipeline proposed in this paper, how can the authenticity of the synthetic EO images be ensured, considering that generative models may introduce artifacts? If the model generates false features, how should they be handled?
>
> Our SAR-to-EO model, like any generative approach, can occasionally produce false features. However, we have observed that increasing the dataset's capacity significantly reduces the occurrence of such artifacts by improving the model's generalization ability.
>
> To further ensure the authenticity of synthetic EO images, methods like those described in [How Faithful is your Synthetic Data? Sample-level Metrics for Evaluating and Auditing Generative Models] [4] can be employed. These approaches provide robust tools to evaluate and mitigate the introduction of false features, helping maintain the reliability of the generated data.
>
>
> [1] Li, B., Xue, K., Liu, B., & Lai, Y. K. (2023). Bbdm: Image-to-image translation with brownian bridge diffusion models. In Proceedings of the IEEE/CVF conference on computer vision and pattern Recognition (pp. 1952-1961).
> [2] Cha, M., Angelides, G., Hamilton, M., Soszynski, A., Swenson, B., Maidel, N., ... & Freeman, B. (2023). MultiEarth 2023--Multimodal Learning for Earth and Environment Workshop and Challenge. arXiv preprint arXiv:2306.04738.
> [3] Rambour, C., Audebert, N., Koeniguer, E., Le Saux, B., Crucianu, M., & Datcu, M. (2020). Flood detection in time series of optical and sar images. The International Archives of the Photogrammetry, Remote Sensing and Spatial Information Sciences, 43(B2), 1343-1346.
> [4] Alaa, A., Van Breugel, B., Saveliev, E. S., & van der Schaar, M. (2022, June). How faithful is your synthetic data? sample-level metrics for evaluating and auditing generative models. In International Conference on Machine Learning (pp. 290-306). PMLR.

---

> > ### Comment · Reviewer_bjda · 2024-11-30
> >
> > I thank the authors for their diligent works. They provide additional detailed information about the dataset in the rebuttal. However, I think the examples in the Application are not so convincing. 1) For GeoChat, this LLM is trained mainly on the RGB images. It also cannot allow the SAR format input. So it is not fair if you change a SAR image into a gray image as an input. 2) For Segmentation,  the authors admit that SAR images outperform SynEO in classes related to natural landscapes. This example does not do justice to the advantages of SynEO images.

---

> ### Author Response · Authors · 2024-11-30
> **Response to reviewer bjda**
>
> We sincerely thank the reviewer for their valuable feedback and for recognizing our efforts in providing detailed information about the dataset. We appreciate the opportunity to clarify the points raised.
>
> ### **Purpose of SynEO and SAR-to-EO Translation:**
>
> Our primary goal with SynEO is not to claim its superiority over SAR images but to highlight its **practical utility**. Specifically, SynEO allows direct compatibility with most EO-based applications (e.g., segmentation, vision-language models) without requiring extensive adaptation. If our comparison with SAR images in the manuscript conveyed a different intention, we apologize for the misunderstanding and **will revise the wording to avoid such confusion.** To emphasize, the purpose of SAR-to-EO translation is not merely a format transformation but to enable the use of pre-existing EO-based models with SAR-derived inputs. To the best of our knowledge, this is the first time this capability has been demonstrated.
>
> ### **Performance on Artificial Structures and Natural Landscapes:**
> While SynEO consistently performs well on artificial structures, we acknowledge the noted differences in performance for natural landscapes. As a benchmark dataset, our goal is to foster further advancements in SAR-to-EO translation techniques. We firmly believe that, with continued development and optimization on our benchmark, it is feasible to achieve strong performance in classes related to natural landscapes as well.
>
> ### **Summary of Contributions:**
> Our primary contribution is the introduction of a **comprehensive benchmark dataset specifically designed for evaluating SAR-to-EO translation**. This dataset enables robust assessment of generalization performance across diverse environments. Additionally, we provide benchmarks covering the entire pipeline, including preprocessing, modeling, and postprocessing, under various settings: paired, unpaired, and partially-paired data. Uniquely, our work extends beyond SAR-to-EO translation by exploring its applications, marking the first effort within the academic community to do so.
>
> Finally, we are deeply grateful for the reviewer’s insights and constructive comments. We are committed to maintaining and updating the benchmark dataset and codebase to support the community’s efforts in this domain.

---

> > ### Author Response · Authors · 2024-12-02
> >
> > We’ve worked so hard to prepare our answers, and now there’s just one day left until the review. We know you’re incredibly busy, but could you please take a moment to look at our responses? We believe they’ll be very helpful for your review. Thank you so much!!!!

---

### Official Review · Reviewer_Pahi · 2024-10-28

**Soundness:** 3
**Presentation:** 2
**Contribution:** 2
**Rating:** 5
**Confidence:** 5

**Summary:**

This paper introduces SAR2Earth, a deep learning benchmark dataset for SAR to optical translation. It verifies the application of this dataset through various methods and perspectives. The content is comprehensive, and both data and code are openly shared.

**Strengths:**

This paper constructs a high-resolution (<1m) SAR to optical translation dataset, covering both rural and urban scenes, including large-scale areas—a first in this field. The quality of the data and methods is commendable, with detailed data processing and sufficient experimental comparison. The introduction is clear, outlining the significance of the dataset, the construction process, and various factors influencing SAR to optical translation. This dataset holds significant potential in the SAR to optical translation domain.

**Weaknesses:**

Although the paper presents a valuable dataset, several shortcomings remain:
1．	The connection between the background introduction and experimental results lacks coherence. The significance of SAR to optical translation is highlighted as solving cloud and fog issues, but the experiments fail to substantiate this claim. While the GeoChat analysis and SAM segmentation results are novel, the GeoChat testing method is questionable and too qualitative. Conversely, the OSM experiment or surface segmentation, which could better demonstrate the conversion's significance, is not included.
2．	The dataset review is not comprehensive, and its positioning is unclear. Since 2018, SAR to optical translation has seen significant advancements with many global datasets such as SEN12MS dataset for Sentinel-2, Planet-CR dataset for Planet, and SMILE dataset for Landsat. These were not mentioned. The main contribution of this dataset lies in its sub-meter resolution, which previous global datasets do not offer, enhancing urban (built-up) scene accuracy.
3．	The experimental discussion is not sufficiently in-depth. The paper discusses speckle noise but does not address the alignment issues with increased resolution using WGS84. The impact of these alignment errors and their solutions should be discussed, which is not encountered in lower-resolution datasets.

**Questions:**

1.	The title mentions datasets and remote sensing applications, yet experimental applications are not clearly reflected. GeoChat and SAM are not true applications. SAR to optical conversion's main significance lies in interpreting scenes in complex weather, which is only mentioned in the background but not experimentally demonstrated.
2.	The dataset review is not comprehensive, and its positioning is unclear. What are the differences and advantages between your dataset and those medium-resolution global datasets?
3.	Is there a better way to deal with the errors of SAR and optical registration, not just WGS84 registration?
4.	Many existing models are compared in the experiments. Can these models handle noise directly within the network without pre-processing before model training?
5.	The time span between SAR and optical data is lengthy, sometimes exceeding a year. How can this limitation be narrowed? Seasonal differences and surface changes may affect the translation model's accuracy. How can this be mitigated?
6.	Can the model trained on this dataset be directly applied in specific scenarios, or how far is SAR to optical translation from real-world application? Which specific scenarios would benefit most?
7.	GeoChat verification seems biased towards optical images. Would describing SAR images show similar biases?
8.	The migration accuracy for urban and rural scenes is low. Does this issue extend to other scenes? Can dataset expansion and method changes address this?

---

> ### Author Response · Authors · 2024-11-21
> **Response to reviewer Pahi**
>
> We sincerely thank the reviewer for recognizing the significance and potential of the SAR2Earth dataset in the SAR-to-optical translation domain. We appreciate the acknowledgment of its high-resolution coverage, detailed data processing, and comprehensive comparisons. The reviewer's feedback motivates us to further refine and enhance our work.
>
> > Weaknesses  & Question1 : The title mentions datasets and remote sensing applications, yet experimental applications are not clearly reflected. GeoChat and SAM are not true applications. SAR to optical conversion's main significance lies in interpreting scenes in complex weather, which is only mentioned in the background but not experimentally demonstrated.
>
> We have added the Visual Grounding task and Land Cover Segmentation to the main manuscript and moved the image interpretation and Segment Anything tasks to the supplementary material. Please refer to **lines 481–527**.
> In our main paper, we define **application** broadly as any practical use of SAR data. Therefore, we consider Visual Grounding, Land Cover Segmentation, GeoChat, and SAM to all be valid applications. The primary significance of our benchmark dataset is to enable SAR data to be used in applications that typically operate with EO data, thereby expanding the usability of SAR.
>
> > Weaknesses  & Question 2 : The dataset review is not comprehensive, and its positioning is unclear. What are the differences and advantages between your dataset and those medium-resolution global datasets?
>
> We appreciate this insightful question. To address it, we have clarified the distinctions between cloud removal and SAR-to-EO translation in **lines 99–107 and Appendix B**. Specifically:
>
> 1. The cloud removal task uses SAR as an auxiliary input to remove clouds from EO imagery. This method relies on multi-temporal EO and SAR data but is constrained by nighttime conditions, when EO imagery is unavailable, and struggles to reconstruct dynamic objects due to temporal discrepancies between SAR and EO acquisitions.
>
> 2. In contrast, the SAR-to-EO translation task generates EO-like images solely from SAR data, allowing operations in both daytime and nighttime conditions without requiring EO imagery. This makes SAR-to-EO translation particularly advantageous in scenarios where EO data is unavailable or cloud-covered.
>
> Furthermore, in the cloud removal task, the SAR image (which is not temporally aligned) serves as an auxiliary input to produce a cloud-free EO image at the EO acquisition time. This limits the ability to consistently generate dynamic objects, as they may be either missing or misaligned between EO and SAR data. However, our SAR-to-EO translation approach generates SynEO imagery at the SAR acquisition time, enabling consistent rendering of dynamic objects based on the SAR reference. As a result, SAR-to-EO translation facilitates the construction of ultra-high-resolution datasets where dynamic objects are fully visible.
>
> We hope these updates address the reviewer’s concerns and provide clarity. Once again, we thank the reviewer for their constructive feedback, which has greatly helped us improve the quality of our work.
>
>
> > Weakness & Question 3 :  The experimental discussion is not sufficiently in-depth. The paper discusses speckle noise but does not address the alignment issues with increased resolution using WGS84. The impact of these alignment errors and their solutions should be discussed, which is not encountered in lower-resolution datasets.
>
> To ensure accurate co-registration between SAR and EO images, we first align their coordinate systems and then perform co-registration using longitude and latitude information. This straightforward approach provides consistency across the entire dataset. However, as resolution increases, factors such as the satellite's imaging angle may affect co-registration accuracy, potentially leading to misalignments.
>
> While feature matching-based methods could address these challenges, **co-registering different modalities like SAR and EO is inherently difficult**. Additionally, such methods often require extensive training and exhibit variable performance depending on the scene.
>
> Given our focus on dataset creation, **we prioritized a basic, method-independent approach that guarantees consistent and reliable performance across diverse regions**. By employing a pixel-matching strategy based on longitude and latitude, we ensure a robust and efficient co-registration method suitable for real-world SAR-to-EO applications. Nevertheless, we believe that enhancing co-registration precision could further improve the robustness of our approach, enabling even better performance in practical scenarios.

---

> ### Author Response · Authors · 2024-11-21
> **Response to reviewer Pahi (2)**
>
> > Question4 : Many existing models are compared in the experiments. Can these models handle noise directly within the network without pre-processing before model training?
>
> Many existing models in the experiments can handle noise directly within the network without pre-processing; however, pre-processing often improves performance. **Table 2** presents results without pre-processing, while **Table 3 compares performance based on different de-noising methods**. As demonstrated in table3 applying pre-processing significantly enhances the performance of SAR-to-EO translation models. Detailed explanations of these experiments, including the settings for the tables, are provided in **Section 5.1 of the paper**. Additional clarifications were added—refer to **Line 340~346** for further details. This highlights that while models are capable of handling noise internally, pre-processing remains a critical factor in optimizing performance.
>
> >  Question 5 : The time span between SAR and optical data is lengthy, sometimes exceeding a year. How can this limitation be narrowed? Seasonal differences and surface changes may affect the translation model's accuracy. How can this be mitigated?
>
> Obtaining SAR and EO data at the same time and location is nearly impossible unless one has direct control over satellite operations. While aligning SAR and EO data more closely in time and space, as demonstrated by datasets like **MAVIC**, can significantly improve model performance, such setups are not feasible under **most real-world conditions** due to the typically lengthy revisit cycles of satellites over the same location. To address this challenge, we **intentionally designed our dataset to include diverse time differences between SAR and EO data**, reflecting real-world scenarios where such time gaps are inevitable. Importantly, our SAR2Earth dataset serves as a benchmark, specifically tailored to real-world SAR-to-EO translation tasks, ensuring its practicality and applicability for real-world use cases.
>
> >  Question 6 :Can the model trained on this dataset be directly applied in specific scenarios, or how far is SAR to optical translation from real-world application? Which specific scenarios would benefit most?
>
> The SAR-to-EO translation task holds significant potential for various scenarios, such as defense, disaster management, and forest monitoring. For instance, deforestation in the Amazon often occurs at night, as do many defense operations. Additionally, disaster response requires rapid action, making SAR an ideal choice due to its ability to operate unaffected by clouds, and both day and night conditions. However, SAR data is notoriously difficult for non-experts to interpret, and training skilled personnel comes with substantial costs. In conclusion, our SAR-to-EO model can be applied to most applications utilizing SAR data, providing a more interpretable and accessible solution for these critical scenarios.
>
> >  Question 7 : GeoChat verification seems biased towards optical images. Would describing SAR images show similar biases?
>
> GeoChat's bias towards optical images is understandable, as it has been predominantly trained on EO data, which is easier to interpret and widely used in similar applications. In contrast, SAR images are significantly more complex and challenging to decode, making it difficult to produce large-scale, high-quality labeled datasets for training language models effectively. This complexity underscores the importance of SAR-to-EO translation as a vital bridge to make SAR data more interpretable and usable for language models. Given the challenges of generating large-scale SAR-specific datasets, we believe that at the current stage, SAR-to-EO translation remains an essential step for incorporating SAR data into such systems.

---

> ### Author Response · Authors · 2024-11-21
> **Response to reviewer Pahi (3)**
>
> > Question 8 : The migration accuracy for urban and rural scenes is low. Does this issue extend to other scenes? Can dataset expansion and method changes address this?
>
> The observed low migration accuracy for urban and rural scenes is indeed a concern, and we believe this issue could extend to other challenging cases as well. **Our dataset serves as a benchmark, and we identify these urban and rural scenes as hard cases due to the significant domain gap between them**. This domain gap—resulting from differences in visual features, density, and spatial composition—leads to decreased performance. However, applying **domain adaptation methods could mitigate these issues by helping the model generalize better across different types of data.** Our benchmark dataset also provides metadata, such as OSM (OpenStreetMap) information and longitude/latitude data, which can further support domain adaptation techniques, ultimately improving the model's robustness. It's important to note that this domain gap was intentional in our benchmark design, as it aims to present a realistic challenge to push the boundaries of model performance in diverse scenarios. By expanding the dataset and adapting the approach, there is considerable potential for performance improvement in these and other challenging scenes.

---

### Official Review · Reviewer_pSfN · 2024-11-01

**Soundness:** 3
**Presentation:** 3
**Contribution:** 3
**Rating:** 8
**Confidence:** 5

**Summary:**

This paper introduces a novel public benchmark dataset called SAR2Earth which designed for SAR-to-EO translation. Both EO and SAR images have inherent limitations: EO images cannot penetrate cloud cover and capture imagery at night; SAR images are affected by speckle noise and complicating analysis. Meanwhile, existing large foundation models and generalization models trained on EO images do not perform effectively on SAR images. As a result, SAR-to-EO translation methods have been proposed to mind this gap. This paper proposes SAR2Earth dataset with co-registered SAR and EO images, which aims to support further research for SAR-to-EO translation. The intuition is the limitation of available SAR-to-EO datasets.
The authors propose a SAR-to-EO pipeline and test the SAR2Earth dataset on various image-to-image translation methods with different preprocessing and postprocessing techniques. The experimental results demonstrate the potential value of this benchmark.

**Strengths:**

•	This paper proposes a comprehensive benchmark dataset called SAR2Earth for SAR-to-EO translation. The SAE2Earth dataset is collected from 8 regions, with rural/urban domains, a wide range of temporal shifts from 1-month to 5-year, and high resolution images ranging from 0.15m to 0.6m.
•	To evaluate the performance of the dataset, this paper proposed a SAR-to-EO pipeline that integrates previous research and provided benchmark results on state-of-the-art image-to-image translation models.
•	The experimental results are comprehensive, detailed, and promising. The proposed public benchmark dataset can support future research in this domain.

**Weaknesses:**

•	The dataset description lacks specific details like the number of images in the dataset and recommended dataset splits.
•	Compared with section 5, the SAR-to-EO pipeline in section 4 occupies only a small portion of the paper and lacks adequate explanation.
•	As the authors point out, this work did not leverage extra modalities, like OSM-based label maps or metadata. These extra information can provide better understanding for regional characteristics in SAR-to-EO translation.

**Questions:**

•	Can you provide more information about the SAR2Earth dataset? E.g. a comparison of the number of images with previous datasets.
•	Can you provide more detail in section 4, e.g. a detailed flow chart? This may provide readers with a comprehensive understanding of the pipeline.

---

> ### Author Response · Authors · 2024-11-21
> **Response to reviewer pSfN**
>
> We sincerely thank the reviewer for their valuable feedback and recognition of the SAR2Earth dataset's novelty and impact. We appreciate the acknowledgment of its comprehensiveness and the clarity of our pipeline and experiments. We will incorporate the insights to improve our work further.
>
> > Weakness 1. The dataset description lacks specific details like the number of images in the dataset and recommended dataset splits.
>
> > Question 1. Can you provide more information about the SAR2Earth dataset? E.g. a comparison of the number of images with previous datasets.
>
> We have added information about the number of images and data splits in the main manuscript. Please refer to **Section L-237~239** and **Appendix A** for these updates.
>
> > Weakness 2. Compared with section 5, the SAR-to-EO pipeline in section 4 occupies only a small portion of the paper and lacks adequate explanation
>
> > Question 2. Can you provide more detail in section 4, e.g. a detailed flow chart? This may provide readers with a comprehensive understanding of the pipeline.
>
> To address this, we have condensed the content in Section 5 and expanded Section 4 to include more details about the refinement step. Specifically, we added a new subsection, **Section 4.3**, which focuses on post-processing. Additionally, to enhance clarity, we have included a detailed flow chart in **Appendix B**. We hope these changes provide a more comprehensive explanation of the pipeline.
>
> > Weakness 3. As the authors point out, this work did not leverage extra modalities, like OSM-based label maps or metadata. These extra information can provide better understanding for regional characteristics in SAR-to-EO translation.
>
> We fully agree with your opinion. Utilizing extra modalities could indeed enhance SAR-to-EO translation performance. To support this potential for other researchers and our future work, we provide OSM-based label maps and metadata, along with a more detailed explanation of the metadata added in the appendix.

---

> > ### Comment · Reviewer_pSfN · 2024-11-27
> > **response for the response**
> >
> > Thank the authors for their reply. I remain my original decision "accept."

---

> > > ### Author Response · Authors · 2024-11-29
> > > **Response to reviewer pSfN**
> > >
> > > Thank you for your insightful feedback! We will finalize our work thoroughly to contribute to the academic community.

---

### Author Response · Authors · 2024-11-21
**General Rebuttal**

General Rebuttal


We appreciate the reviewers' thoughtful feedback and insightful suggestions, which guided us in refining our work. In response, we made several significant updates to the manuscript, which are summarized below.


1. We have updated **Section 3.2** and added **Appendix A** to include more detailed information about the SAR2Earth dataset.


2. We have added comparisons with cloud removal datasets in **Section 2.2 and Appendix B** to highlight the unique features of SAR2Earth.


3. Additional qualitative results have been added to **Figure 4**, and a detailed pipeline description has been updated in **Section 4.3 and Appendix B**.


4. New experimental results for Land Cover Segmentation and Visual Grounding have been introduced and updated in **Sections 5.1 and 5.6**.


For a detailed discussion of the changes, please refer to the specific responses to individual comments. Thank you again for helping us enhance the quality of this work.

---

### Author Response · Authors · 2024-12-04
**Summary of Revision and Discussion**

Dear Program Chairs (PC), Senior Area Chairs (SAC), Area Chairs (AC), and Reviewers,

As the discussion phase concludes, we would like to express our sincere gratitude to all participants for their time and valuable insights. The thoughtful questions and feedback provided have been instrumental, and we have worked diligently to address the concerns raised.

We enhanced the dataset description with additional details on image counts, data splits, and regional diversity for greater clarity. Additionally, we introduced new experiments, such as Land Cover Segmentation and Visual Grounding, to demonstrate the effectiveness of SAR-to-EO translation. Finally, we emphasized SAR2Earth’s role as a benchmark dataset to advance SAR-to-EO technologies and support real-world applications and future research.

We are pleased that our responses addressed key concerns raised by the reviewers, resulting in positive feedback and improved scores. While challenges such as time discrepancies and high-resolution SAR-EO co-registration remain, our approach lays a strong foundation for further exploration in these areas.

The recognition of the dataset’s high-resolution data and challenging benchmarks is greatly appreciated. While reviewers Pahi and LqKw suggested it might fit better in another venue, we emphasize that the focus on "datasets and benchmarks" aligns closely with the objectives of this community. The constructive feedback provided has been invaluable, and we deeply appreciate the effort invested in reviewing this work.

We hope the revisions and explanations will be carefully considered during the final evaluations. Once again, our sincere thanks go to all reviewers for their contributions to improving this work.

Best regards,

The Authors

---

### Meta-Review · Area_Chair_KS7s · 2024-12-23

**Metareview:**

This paper introduces a new dataset, SAR2Earth, designed to benchmark SAR-to-EO image-to-image translation methods. The dataset consists of paired SAR and EO image patches extracted from 18 spatially aligned pairs of high-resolution (<1m) SAR and EO images, collected from 8 distinct regions spanning both urban and rural areas. The paper benchmarks several paired and unpaired methods for SAR-to-EO image-to-image translation, alongside SAR image denoising. While the primary focus of the paper is image-to-image translation, it also addresses image denoising as part of the pipeline, which has relevance for certain downstream applications.

Reviewers have recognized the dataset as a valuable resource, given its provision of high-resolution (<1m) SAR and EO image pairs. These pairs are not only beneficial for image-to-image translation but also for other tasks that require learning cross-modal representations in remote sensing.

However, the paper's scope appears somewhat diffuse. The benchmarking covers denoising, image-to-image translation, and other applications, which detracts from a more focused presentation on the dataset and the benchmarking of paired and unpaired image-to-image translation methods. Additionally, several SOTA methods that are highly relevant to this task, such as BicycleGAN, MUNIT, NICE-GAN, and Attn-CycleGAN, are not considered for the benchmark.

There are other issues in the paper that remain unclear. For instance, Table 1 states that SAR2Earth's sensor type is "Google Earth." If Google Earth was used to collect Sentinel-2 images, as suggested by the project's website (https://sar2earth.github.io/), it raises concerns about the feasibility of obtaining EO images at the same high resolution as Capella SAR images. Additionally, the temporal misalignment or discrepancies between SAR and EO images could lead to inaccuracies in representation learning. Moreover, the lack of SAR-EO co-registration is a significant issue for high-resolution paired images, especially for tasks like image-to-image translation, where accurate co-alignment is important.

**Additional Comments On Reviewer Discussion:**

The authors made commendable efforts to address the reviewers' comments during the rebuttal phase, leading to improved scores from some reviewers. However, reviewer Pahi remained highly critical of the dataset, particularly regarding the temporal discrepancies and the lack of SAR-EO co-registration. These issues still need to be adequately addressed by the authors.

---

### Decision · Program_Chairs · 2025-01-22

Reject